# Identifying surface reaction intermediates with photoemission tomography

Xiaosheng Yang [1,2,3], Larissa Egger[4], Philipp Hurdax[4], Hendrik Kaser [5], Daniel Lüftner [4], François C. Bocquet [1,2], Georg Koller [4], Alexander Gottwald [5], Petra Tegeder [6], Mathias Richter [5], Michael G. Ramsey[4], Peter Puschnig [4], Serguei Soubatch [1,2] & F. Stefan Tautz [1,2,3]

The determination of reaction pathways and the identification of reaction intermediates are key issues in chemistry. Surface reactions are particularly challenging, since many methods of analytical chemistry are inapplicable at surfaces. Recently, atomic force microscopy has been employed to identify surface reaction intermediates. While providing an excellent insight into the molecular backbone structure, atomic force microscopy is less conclusive about the molecular periphery, where adsorbates tend to react with the substrate. Here we show that photoemission tomography is extremely sensitive to the character of the frontier orbitals. Specifically, hydrogen abstraction at the molecular periphery is easily detected, and the precise nature of the reaction intermediates can be determined. This is illustrated with the thermally induced reaction of dibromo-bianthracene to graphene which is shown to proceed via a fully hydrogenated bisanthene intermediate. We anticipate that photoemission tomography will become a powerful companion to other techniques in the study of surface reaction pathways.

[1] Peter Grünberg Institut (PGI-3), Forschungszentrum Jülich, 52425 Jülich, Germany. [2] Jülich Aachen Research Alliance (JARA), Fundamentals of Future Information Technology, 52425 Jülich, Germany. [3] Experimental Physics IV A, RWTH Aachen University, 52074 Aachen, Germany. [4] Institute of Physics, University of Graz, NAWI Graz, 8010 Graz, Austria. [5] Physikalisch-Technische Bundesanstalt (PTB), 10587 Berlin, Germany. [6] Physikalisch-Chemisches Institut, Ruprecht-Karls-Universität Heidelberg, 69120 Heidelberg, Germany. Correspondence and requests for materials should be addressed to S.S. (email: s.subach@fz-juelich.de)

Since synthetic organic chemistry emerged in the mid-nineteenth century[1], the necessity to correctly identify reaction intermediates and products has led to many techniques and diverse approaches, ranging from classical methods of wet chemistry, including colour reagents and chromatography, to modern variants of nuclear magnetic resonance spectroscopy, to name only a few examples. The applicability of these methods in surface chemistry is often restricted by the specific situation that reactants, intermediates and products remain surface-adsorbed throughout the reaction. This requires the use of surface analytics. But even the rich arsenal of surface science techniques, based for instance on mass spectrometry[2–4], vibrational[5,6] or core level spectroscopies[7–9] and scanning probe microscopies[10–20], does not always allow the authoritative identification of an adsorbed species. For example, it is often difficult to identify the degree of (de-)hydrogenation of a given molecule, as hydrogen is difficult to trace by surface sensitive techniques. Only indirect conclusions can be made from, e.g., the presence or absence of specific vibrational modes in infrared spectroscopy or chemical shifts in core level spectroscopy. Often, however, these data are ambiguous and leave room for interpretation. In contrast, orbitals of molecules, being extremely sensitive to the composition and chemical structure, should allow for unambiguous identification of chemical species.

From this perspective, the momentum-space orbital imaging based on angle-resolved ultraviolet photoemission spectroscopy (ARUPS) is promising. The method of photoemission tomography (PT) developed in the last few years[21–26] is a combined experimental and theoretical technique in which the results of ARUPS are interpreted in terms of the molecular orbital structure of the electron's initial state. In a nutshell, a set of maps of the photoelectron intensity distribution in momentum space (k-maps), recorded at energies corresponding to specific molecular emissions, does not only provide a unique fingerprint of the adsorbed species, but allows a straightforward determination of its precise structure. As we will demonstrate, photoemission tomography can thus be employed for the identification of reaction intermediates and can therefore be useful for disentangling surface reaction pathways.

The polymerisation of 10,10′-dibromo-9,9′-bianthracene (DBBA, $C_{28}H_{16}Br_2$) on metal surfaces is used for the on-surface synthesis of graphene nanoribbons[27–31] and graphene[30,32,33]. The reaction is thermally activated and proceeds via dehalogenation and cyclodehydrogenation. On Cu(110), the enhanced chemical reactivity of the surface prevents polymerisation at moderate temperatures (250 °C)[32] resulting in a well-defined reaction intermediate, which at higher temperature (>750 °C) transforms into graphene. Based on scanning tunnelling microscopy (STM) and x-ray photoemission spectroscopy (XPS), combined with adsorption energy calculations, a partly dehydrogenated state of bisanthene (Fig. 1) has been suggested[32], but the precise nature of this intermediate is debated. The surface reaction of DBBA on Cu

(110) is therefore an excellent test case for PT and its ability to identify the exact chemical state of an intermediate of a surface chemical reaction.

Here we apply photoemission tomography to demonstrate the thermally induced chemical reaction of DBBA on Cu(110) and reveal the exact chemical state of the reaction intermediate. We consider three possible reaction intermediates, which only differ in the degree of hydrogenation at the zig-zag edges of the molecule. When comparing their density of states computed within density functional theory with experimental energy distribution curves from ultraviolet photoemission spectroscopy, an assignment of the intermediate cannot be made. Only when taking into account the angular distribution of the photoemitted electrons from the frontier orbitals, both in the experiment and in the simulation, are we able to unambiguously identify the exact chemical state of the intermediate as a chemisorbed, but fully hydrogenated bisanthene (phenanthro(1,10,9,8-opqra)perylene, $C_{28}H_{14}$) molecule. This demonstrates the outstanding potential of PT in recognising the precise chemical state of molecular adsorbates, especially as far as the chemistry of the molecular periphery is concerned.

## Results

**Density of states**. We have deposited DBBA (Fig. 1) on clean Cu(110) at room temperature and, as a first step, investigated the electronic structure of the as-deposited molecules by conventional ARUPS. Energy distribution curves (EDCs) in the valence band range between the Fermi level and the onset of the Cu d-band emissions are shown in Fig. 2a. At least two adsorbate-related features are discernible, at ~0.15 eV and between 1.0 and 1.5 eV. However, the precise binding energies of these broad features are difficult to determine. Moreover, the anisotropy between the EDCs recorded at the two principal azimuths of the substrate is much weaker than for other organic monolayers on metal (110) surfaces[21,22,34]. Nevertheless, we will show below that the angular distribution of photoelectrons exhibits a twofold symmetry and thus points to a well-defined orientation of the as-deposited species.

After annealing the sample at 250 °C, the EDCs are clearly modified (Fig. 2b). We now find three molecular emissions, one of them almost completely concealed by the rising flank of the Cu d-electrons. The other two appear as well-defined peaks at 0.5 and 1.15 eV. The anisotropy between the two azimuths is now pronounced, suggesting a high degree of molecular orientation after annealing.

**Momentum maps**. On the basis of the experimental EDCs, the nature of the molecular emissions, both before and after annealing, remains unclear. We have therefore measured angular photoelectron distributions at the binding energies in question (see Methods, Supplementary Methods and Supplementary Fig. 1). Corresponding momentum maps (or k-maps), which can also be viewed as a momentum-space resolved density of states (DOS), are displayed for the as-deposited molecules at binding energies of 0.15, 0.9 and 1.4 eV (Fig. 3a), and for the annealed molecules at 0.5, 1.15 and 1.9 eV (Fig. 3b).

For the as-deposited species we observe diffuse patterns that are superimposed on sharp features from Cu sp-band emissions (Fig. 3a). These patterns are twofold symmetric and clearly vary with binding energy, indicating that they must originate from well-defined and distinct molecular orbitals, in spite of the indistinct density of states in Fig. 2a. Both the diffuseness of the momentum maps and the indistinctiveness of the EDCs indicate the co-existence of more than one configuration of the

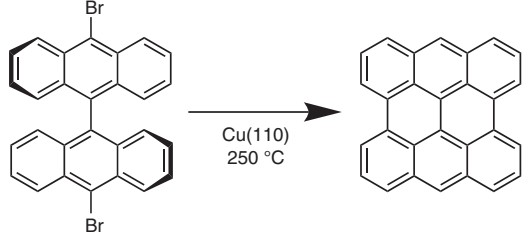

**Fig. 1** Surface reaction of 10,10′-dibromo-9,9′-bianthracene (DBBA). Dehalogenation and cyclodehydrogenation of DBBA on Cu(110) resulting in bisanthene

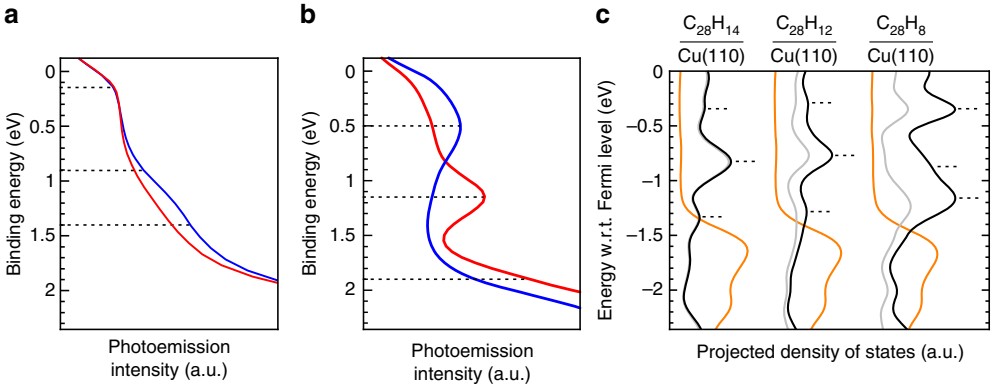

**Fig. 2** Valence band of DBBA on Cu(110) before and after annealing at 250 °C. **a** Energy distribution curves (EDCs) of as-deposited DBBA measured along [001] (blue curve) and [$\bar{1}$10] (red curve) directions of Cu(110). **b** EDCs after annealing of DBBA at 250 °C measured along [001] direction of Cu(110) (blue curve) and 25° away from [$\bar{1}$10] direction (red curve). For the EDCs, the photoelectron intensity was integrated over the polar angle range of 0° to +85°. **c** DFT calculations of density of states (DOS) of $C_{28}H_{14}$, $C_{28}H_{12}$ and $C_{28}H_8$ on Cu(110). The orange lines show the total DOS, while the black and grey lines display the DOS projected onto the molecule and the $\pi$-states of the molecule, respectively. The dotted lines mark energies corresponding to experimental and theoretical $k$-maps in Figs. 3 and 4

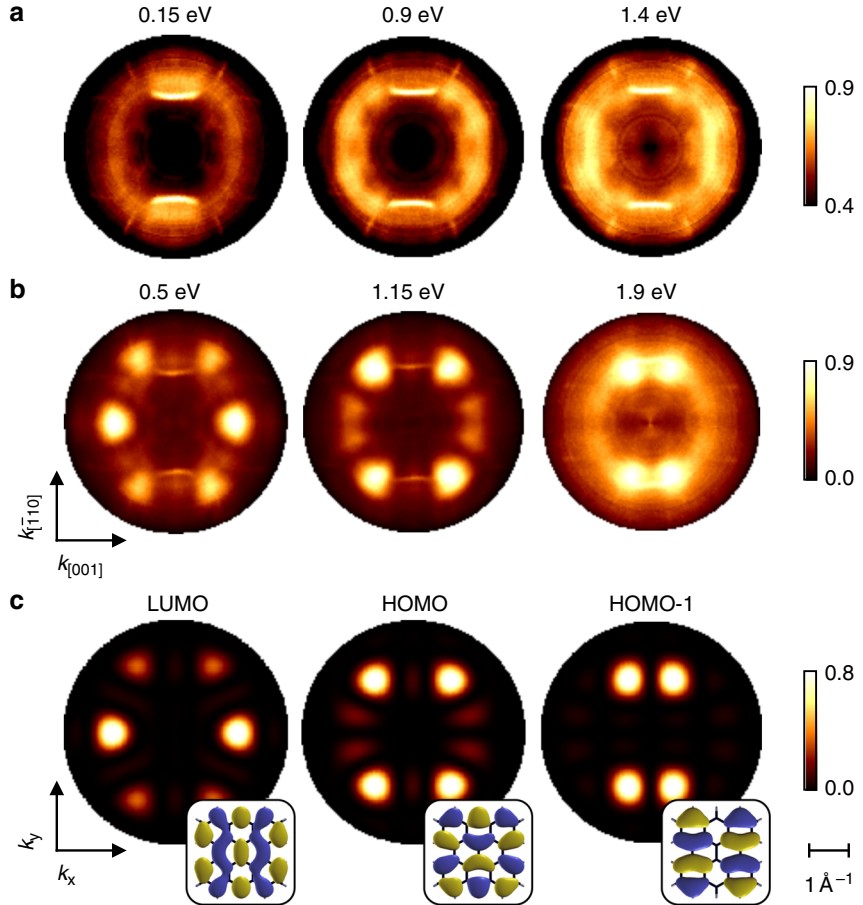

**Fig. 3** Photoelectron distributions. Experimental $k$-maps of DBBA **a** before and **b** after annealing measured at corresponding binding energies (compare dotted lines in Fig. 2a, b). **c** Theoretical $k$-maps of LUMO, HOMO and HOMO-1 orbitals of a bisanthene. Corresponding orbitals are shown in insets

adsorbed species (see Supplementary Note 1, Supplementary Table 1 and Supplementary Figs. 2–4).

After annealing to 250 °C, the momentum maps appear more defined (Fig. 3b). In agreement with the EDCs in Fig. 2b, this suggests a higher degree of orientation and indicates the presence of only one surface species. Most importantly, however, the changes in the momentum maps prove a significant modification

of the orbital structure, indicating that a thermally activated surface reaction to a new species has occurred.

A plausible candidate for this reaction intermediate is bisanthene, which has already been introduced in Fig. 1. We have computed the gas phase electronic structure of bisanthene and simulated the expected angular photoelectron distributions by calculating the Fourier transform of its molecular orbitals[21]

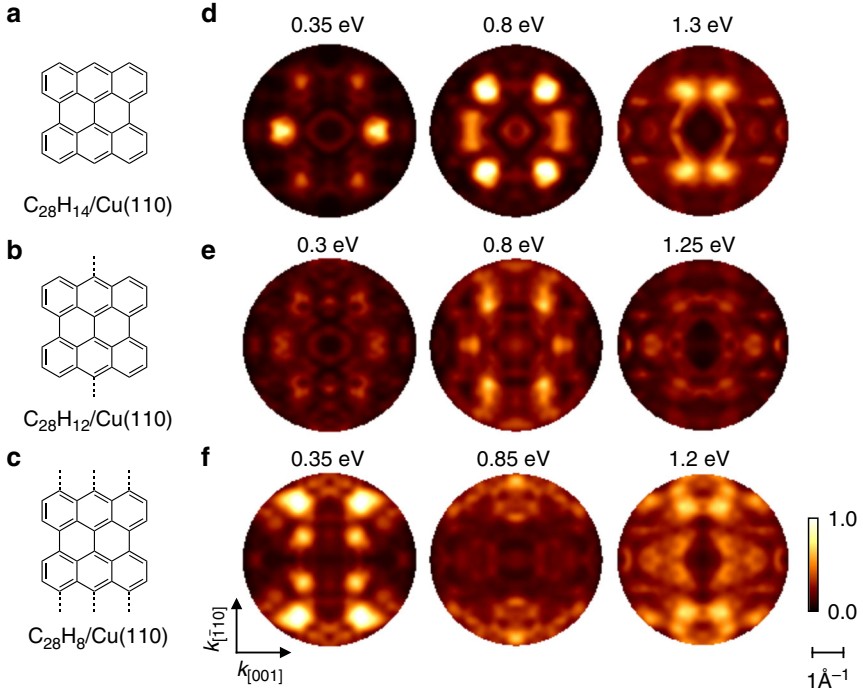

**Fig. 4** Bisanthene, $C_{28}H_{12}$ and $C_{28}H_8$ on the Cu(110) surface. **a–c** Chemical structure and **d–f** simulated $k$-maps at three representative binding energies (compare dotted lines in Fig. 3c) for **a**, **d** bisanthene $C_{28}H_{14}$ and partly dehydrogenated **b**, **e** $C_{28}H_{12}$ and **c**, **f** $C_{28}H_8$. Grey dashed lines in **b**, **c** mark localised chemical bonds of dehydrogenated carbon atom with copper atoms

(see the Methods section). The resulting momentum maps of the lowest unoccupied molecular orbital (LUMO), the highest occupied molecular orbital (HOMO) and the HOMO-1 are depicted in Fig. 3c. They match very well the three measured patterns at 0.5, 1.15 and 1.9 eV, respectively, and thereby give strong evidence that during annealing at 250 °C on Cu(110), DBBA undergoes dehalogenation and cyclodehydrogenation reactions during which the adsorbate planarizes and the $\pi$-conjugation expands to the entire fused carbon backbone. The momentum maps in Fig. 3b, c further reveal that, first, bisanthene is oriented along the $[\bar{1}10]$ direction of the Cu(110) surface, and second, that bisanthene chemisorbs on Cu(110). The latter is proven by the charge transfer into the formerly unoccupied LUMO, which makes this orbital observable in PT.

**Reaction intermediate**. Although the agreement between angular photoelectron distributions of gas phase bisanthene and the measured patterns is good, there are deviations, which could in principle point to small modifications of bisanthene, such as (partial) dehydrogenation. Therefore, the exact chemical state of the reaction intermediate is not yet clear.

Focusing on the possibility of edge dehydrogenation, there are in fact two plausible suspects in addition to bisanthene ($C_{28}H_{14}$, Fig. 4a). These are $C_{28}H_{12}$ (Fig. 4b) and $C_{28}H_8$ (Fig. 4c). The latter two would form covalent bonds with the Cu surface through dehydrogenated C atoms. Based on STM/XPS and density functional theory (DFT) $C_{28}H_8$ has in fact been suggested as the correct reaction intermediate[32] (cf. Supplementary Notes 3 and 4, Supplementary Figs. 8 and 9 for discussion).

To shed light on the true nature of the reaction intermediate, we first compare the density of states for the adsorbed $C_{28}H_{14}$, $C_{28}H_{12}$ and $C_{28}H_8$ as calculated by DFT and projected onto the molecule (pDOS). At the level of van der Waals-corrected DFT, their relaxed adsorption structures are as follows: bisanthene occupies the short-bridge site on Cu(110), while $C_{28}H_{12}$ and $C_{28}H_8$ prefer on-top sites. In all three cases the molecules are

oriented with their central C-C bond along the $[\bar{1}10]$ direction of the Cu(110) surface (cf. Supplementary Note 2, Supplementary Table 2 and Supplementary Figs. 5–7 for details). Although the pDOS shown in Fig. 2c contains peaks which can be associated with the states observed in the experimental EDC in Fig. 2b, there is, however, no clear fingerprint which could be used to unambiguously identify the exact nature of the reaction intermediate. Yet, it is interesting to note that the contribution of $\pi$-states to the total pDOS gradually decreases from intact bisanthene to $C_{28}H_8$ (Fig. 2c). This is caused by the strong concave distortion of the dehydrogenated bisanthene derivatives, resulting from chemical bonds that form between unsaturated edge carbons and the Cu(110) surface[33].

It is reasonable to assume that the strong molecular distortion apparent in the pDOS will modify the corresponding orbital structure significantly. We thus turn again to momentum maps, since these are a direct representation of the orbital structure. Evidently, we need to consider the patterns of the combined adsorbate/substrate system including the local bonds at the edges in order to determine the precise chemical nature of the surface intermediate. We employ here a recently developed extension of PT in which the momentum map simulations are no longer based on gas phase molecular orbitals, but on the wave functions of the (strongly) interacting molecule/metal interface[35].

The results of these simulations are depicted in Fig. 4d–f for $C_{28}H_{14}$/Cu(110), $C_{28}H_{12}$/Cu(110) and $C_{28}H_8$/Cu(110), respectively. For each chemical species, the simulated momentum maps are shown at three representative binding energies, as indicated by the dotted lines in the pDOS curves of Fig. 2c. We notice profound differences in the momentum maps for the three species, demonstrating a striking sensitivity of PT to the hydrogen saturation of molecular edges. Dehydrogenation alters the patterns and moreover leads to increasingly more diffuse emissions signatures, most likely because the local bonds of the dehydrogenated carbon atoms with the Cu(110) mix substrate states into the molecular wave functions.

The comparison of the simulated momentum maps with the experimental ones (Fig. 3b) allows us to exclude both dehydrogenated species $C_{28}H_{12}$ and $C_{28}H_8$. We conclude that the fully hydrogenated bisanthene is the sought-after reaction intermediate. The close resemblance of the gas phase simulations (Fig. 3c) with the momentum maps of the adsorbed bisanthene (Fig. 4d) demonstrates that its orbital structure suffers only minor changes upon adsorption, despite the former LUMO being filled and involved in the bonding of the molecule to the metal. A similar robustness of molecular orbitals has been demonstrated for a number of $\pi$-bonded systems[21–23,35].

**Reaction energies.** Our identification of fully hydrogenated bisanthene as the reaction intermediate is in contrast to Simonov et al. who have identified the dehydrogenated $C_{28}H_8$ as this intermediate[32]. To resolve this disagreement, we calculate the energetics of forming the three possible intermediates. The adsorption energy, defined as the total energy difference between the combined system and its constituents, turns out to be $-5.3$, $-8.0$ and $-12.7$ eV for $C_{28}H_{14}$, $C_{28}H_{12}$ and $C_{28}H_8$, respectively (see Supplementary Note 2 and Supplementary Table 2), and would therefore indicate $C_{28}H_8$ as the most stable intermediate in agreement with Simonov et al. However, this adsorption energy disregards the energy that is necessary to split the carbon-hydrogen bonds. When computing the full chemical reaction energies for the three considered intermediates,

$$C_{28}H_{16}Br_2/Cu(110) \begin{cases} \to C_{28}H_{14}/Cu(110) + 2Br/Cu(110) + 2H/Cu(110) & (-6.8\,\text{eV}) \\ \to C_{28}H_{12}/Cu(110) + 2Br/Cu(110) + 4H/Cu(110) & (-4.1\,\text{eV}) \\ \to C_{28}H_8/Cu(110) + 2Br/Cu(110) + 8H/Cu(110) & (+2.0\,\text{eV}) \end{cases}$$
$$(1)$$

we obtain reaction energies (in parenthesis in Eq. (1)) which demonstrate that the fully hydrogenated bisanthene ($C_{28}H_{14}$) is the most energetically favourable intermediate in agreement with PT.

## Discussion

The ability of PT to image orbitals makes it particularly sensitive to the exact chemical state of surface reaction species. We demonstrated this for the thermally activated DBBA on Cu(110), where the fully hydrogenated bisanthene ($C_{28}H_{14}$), with its LUMO occupied by charge transfer from the surface, is identified as the reaction intermediate prior to the formation of graphene at higher annealing temperatures. In contrast to other experimental techniques, PT can differentiate the chemistry of the molecular periphery, i.e., the degree of (de-)hydrogenation, as well as determine the occupation of the frontier orbitals. As photoemission tomography is neither constrained by the need for cryogenic temperatures nor to planar surface species, we envision it to be a complementary companion to atomic force microscopy and other state-of-the-art surface science methods in the study of reaction pathways at surfaces. Its major restrictions are those of vacuum and conductive substrates inherent to ARUPS.

## Methods

**Sample preparation.** Sample preparation and ARUPS experiments have been carried out in ultra-high vacuum with a system base pressure in the $10^{-10}$ mbar range. The single crystal Cu(110) was cleaned by repeated cycles of $Ar^+$ ion sputtering at 1000 eV followed by annealing at 800 K. 10,10′-dibromo-9,9′-bianthracene (Sigma-Aldrich, CAS number 121848-75-7) was evaporated from a Knudsen type molecular evaporator (Kentax GmbH) at 185 °C onto the Cu(110) surface kept at room temperature (RT). The molecular flux was calibrated using a quartz microbalance to obtain the nominal thickness of the DBBA film of one monolayer. All photoemission experiments were conducted at RT for the cases of as-deposited DBBA/Cu(110) and after several steps of annealing ranging between 250 and 750 °C.

**Photoemission experiments.** Photoemission was excited with $p$-polarised light with an incidence angle of 40°, at the Metrology Light Source insertion device beamline[36] of the Physikalisch-Technische Bundesanstalt (Berlin, Germany). A photon energy of 35 eV was used, which corresponds to the highest photoemission cross-section in the present experiment. Photoemission data was recorded with the toroidal electron analyser[37] over an emission angle range from $-85°$ to $+85°$ (for details see Supplementary Methods). For the presented energy distribution curves, the photoelectron intensity was integrated over the polar angle range of 0° to $+85°$ in selected azimuthal directions. Momentum maps were recorded by measuring the photoelectron intensity in the positive polar angle range for a chosen binding energy and rotating the sample in the azimuthal direction in 1° steps. This results in the full photoelectron distribution in the ($k_x$, $k_y$) plane perpendicular to the sample normal.

**Density functional theory calculation.** The electronic structure calculations and the simulations of the momentum maps are based on ab initio computations within the framework of density functional theory. We have performed two types of calculations. First, for the isolated (gas phase) bisanthene molecule ($C_{28}H_{14}$) the NWChem code has been utilised[38], and second, for the full molecule/Cu(110) adsorbate system the VASP code was used[39,40].

For the calculations of the isolated bisanthene molecule, we employ the generalised gradient approximation (GGA)[41] for exchange-correlation effects. The momentum maps of the LUMO, HOMO and HOMO-1 are obtained as Fourier transforms of the respective Kohn–Sham orbitals[21].

The electronic structure calculations for DBBA monolayers adsorbed on Cu(110) and monolayers of $C_{28}H_x$ on Cu(110) are performed by a repeated slab approach by using epitaxial matrices of $\begin{pmatrix} 4 & 0 \\ 1 & 6 \end{pmatrix}$ and $\begin{pmatrix} 4 & 0 \\ 2 & 5 \end{pmatrix}$, respectively. Using a lattice parameter of $a_{Cu} = 3.61$ Å, the metallic substrate is modelled by five Cu layers where a vacuum layer of at least 15 Å has been added between the slabs to avoid spurious electrical fields[42]. For each molecule/metal interface, the most favourable adsorption site is determined by testing the four high-symmetry adsorption sites (hollow, top, short bridge and long bridge) and performing local geometry optimisations in which the two topmost Cu-layers and all molecular degrees of freedom are allowed to relax until forces were below 0.01 eV Å$^{-1}$. The GGA[41] is used and van-der-Waals corrections according to the Tkatchenko-Scheffler method are added[43,44]. Using the projector augmented wave (PAW) method[45], a plane-wave cutoff of 500 eV is employed. For $k$-point sampling, a Monkhorst-Pack grid of $4 \times 4 \times 1$ points is used and a first-order Methfessel-Paxton smearing of 0.1 eV is utilised. Based on the relaxed adsorption geometries, we have computed (projected) densities of states and simulated momentum maps by assuming a damped plane wave as final state as described in more detail in a recent publication[35].

## Data availability

On request, the experimental data are available from S.S. and the theoretical data from P.P.

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

## Acknowledgements

Financial support from the Austrian Science Fund (FWF) (projects P27649-N20, P27427-N20, I3731) and the Deutsche Forschungsgemeinschaft (DFG) (project Ri 804/8-1 and through the SFB 1083 "Structure and Dynamics of Internal Interfaces", project A12) is gratefully acknowledged. We thank John Riley (La Trobe University, Australia) for experimental support.

## Author contributions

F.C.B., P.T., M.R., M.G.R., P.P., S.S. and F.S.T. conceived the research. X.Y., L.E., P.H., H.K., G.K., A.G., M.G.R. and S.S. carried out the experiments. X.Y. and S.S. analysed the data. D.L. and P.P. carried out DFT calculations. M.G.R., P.P., S.S. and F.S.T. wrote the paper with contributions from all authors.

## Additional information

**Competing interests:** The authors declare no competing interests.

