## [Peer Review File · Nature Communications]

Reviewers' Comments:

Reviewer #1:

Remarks to the Author:

The present paper shows beautiful tomographic images of photoelectrons emitted from molecules on the Cu(110) surface. The experimental method itself has been well-known to identify a chemical specie from shapes of the molecular orbitals. Thus, I do not find any novelty for an article of Nature Communications. On the other hand, I find originality of the research paper in identifying reaction intermediates during chemical growth of graphene layers on a Cu substrate. Then, the main claim by the authors is that the intermediate reported by Simonov et al. [32] is wrong. In contrast to broad readers of Nature Communications, this focus of the paper is quite specific. Thus, the paper should be considered in much specialized journal that deals the layer growth. My comments are following:

- 1) To capture interests of broad readers, it is necessary to elaborate significance of the current finding in industrial production of graphene layers.
- 2) Consistencies are not well-described between the present research and the report by Simonov et al [32].
 - i) The $C_{28}H_8/Cu(110)$ model by Simonov et al [32] explains three components in the C 1s core-level spectrum, shown in Fig.6 (b) in [32]. But the $C_{28}H_{14}/Cu(110)$ model cannot explain the experimental evidence since it should have only two components. Thus, the model sounds incorrect.
 - ii) In Fig.7 of [32], Simonov et al directly observed by STM the molecular orbital in real space that matches with $C_{28}H_{14}$ in vacuum and $C_{28}H_8/Cu(110)$. This indicates that shapes of the molecular orbitals are similar between $C_{28}H_{14}$ and $C_{28}H_8/Cu(110)$. On the other hand, simulated patterns of photoemission tomography are very different between molecules, as shown in Fig.4 in the present manuscript. To make the direct and fair comparisons with [32], images of the molecular orbitals must be added individually for nine tomography patterns in Fig.4.
- 3) The authors chose a photon energy of 35 eV for the experiment. Photon- energy dependence of the photoemission tomography should be mentioned. If the tomographic patterns vary with photon energy, choice of 35 eV should be described.

Reviewer #2:

Remarks to the Author:

The present work by Yang et al. investigates the thermal induced reaction of DBBA on Cu(110) to graphene. The authors apply experimental and theoretical methods including photoemission tomography (PT) to identify a reaction intermediate that was previously possibly misinterpreted. The manuscript is well written, the findings are convincing and well presented, and the obtained results are novel and demonstrate the potential of PT to investigate dehydrogenation reactions that are often difficult to explore using other techniques. Therefore I believe that the present work is of great interest to the community of theoretical and experimental surface scientists and deserves being published in nature communications.

I have only two points that should be addressed before publication:

-Why have the (orbital-)energies of the isolated molecule been calculated using the HSE06 xc-functional (Fig. 3c), whereas the band structure of the adsorbed molecules have been computed using the PBE functional (Fig. 4 d-f)? This inconsistency makes the claimed resemblance between Fig.3c and 4d less convincing and I would recommend to use the same functional in all systems.

-In the abstract the authors say "we show that PT is extremely sensitive to the exact wave function". The present work employs KS-DFT orbitals to calculate momentum maps that are compared to experimental maps. The term "exact wave function" is not appropriate and should be modified.

Reviewer #3:

Remarks to the Author:

In their manuscript entitled 'Identifying surface reaction intermediates with photoemission tomography' the authors describe the comparison of experimental and simulated k-space patterns of molecular states as a method to identify reaction intermediates in a surface-chemical reaction. The manuscript addresses the surface-assisted activation of dibromobianthryl (DBBA), which has a significant relevance in the field of bottom-up fabrication of graphene nanoribbons. The photoemission tomography data shown in the work is of high quality and is accompanied by a convincing simulation based on molecular conformations modeled by DFT. While the presented results give convincing new insights on a particular reaction step the chosen system (DBBA on Cu(110)), it also becomes clear that applicability of the method is limited to a very narrow selection of molecule-substrate combinations. For instance, DBBA was designed for yield 7-atom wide graphene nanoribbons on Au(111). This nanoribbons cannot be achieved on Cu(110) due to a fundamentally different reaction path on this surface. I am therefore afraid that the work cannot reach the wide readership at which Nature Communications is aiming. The insights gained here are from both sides, the presented method and the reaction investigated, too specific and more suitable for a specialized journal focusing on, for instance, surface-analytical aspects.

On this basis I cannot recommend the manuscript for publication in Nature Communications.

The authors also should consider the following points:

- The current discussion of the reaction mechanism assumes formation of bisanthrene as an intermediate step. It is, however, not described how hydrogen passivation after dehalogenation would occur.

- The angle under which the EDCs shown in Fig. 2 were measured should be indicated.

- The caption of Fig. 2 does not specify the temperature to which the sample has been annealed.

- Currently, the conditions for the applicability of the method are indicated only in a very indirect way (unique orientational alignment fo adsorbates). This should be improved.

Reviewer #1 (Remarks to the Author):

The present paper shows beautiful tomographic images of photoelectrons emitted from molecules on the Cu(110) surface. The experimental method itself has been well-known to identify a chemical specie from shapes of the molecular orbitals. Thus, I do not find any novelty for an article of Nature Communications. On the other hand, I find originality of the research paper in identifying reaction intermediates during chemical growth of graphene layers on a Cu substrate.

Authors reply:

We thank the Reviewer for their appraisal of the quality of our experimental results. However, we cannot agree with their conclusion about the novelty of our manuscript. To the best of our knowledge, *orbital imaging* has never been used so far for identification of the reaction intermediates or products. This is similar to several publications on intermediates identification with scanning tunneling microscopy (STM) that appeared recently in high-impact journals including the Nature family (cf. references in our manuscript), despite the fact that STM is known since 1981. Moreover, orbital imaging has not been used so far for identification of the exact chemical state of a molecule. Conventionally, core-level spectroscopy (x-ray photoelectron spectroscopy, XPS) is applied for this purpose (as noted by the Reviewer themselves, cf. below) using chemical shifts of particular core levels (as noted in the introductory part of our manuscript). We report (in our manuscript and in the present Reply, cf. below) that the orbital imaging of photoemission tomography is more sensitive to the exact chemical state of reaction intermediates than XPS.

Then, the main claim by the authors is that the intermediate reported by Simonov et al. [32] is wrong. In contrast to broad readers of Nature Communications, this focus of the paper is quite specific. Thus, the paper should be considered in much specialized journal that deals the layer growth.

Authors reply:

We do not understand how the Reviewer could be under this misapprehension. Our aim is not to disprove the conclusions of Simonov et al. (ACS Nano 9 (2015) 8997), but to present photoemission tomography as a method to identify the precise chemical states of compounds involved in surface chemical reactions, especially intermediates. This is explicitly formulated in the title, the introductory part and the concluding discussion. This method can be successfully applied in research and industry and thus must be of interest for the community of chemists, physicists and material scientists.

The DBBA reaction on Cu(110) reported by Simonov et al. was chosen only as an example demonstrating the benefits of using photoemission tomography. In our manuscript we deliberately bypass the discussion of Simonov's paper and their interpretation of XPS and STM results, thereby avoiding sending the reader the wrong message regarding the scope of our work.

Surprisingly, we found that despite the Reviewer's own words "***The experimental method itself has been well-known to identify a chemical specie from shapes of the molecular orbitals***", they invoked the XPS and STM results of Simonov et al. to question our interpretation. In response, we include now a short discussion of this in the present Reply (cf. below) and in the

Supplementary Information. This reveals that there are cases where traditional methods (XPS and STM) cannot guarantee the correct identification of the chemical states.

My comments are following:

1) To capture interests of broad readers, it is necessary to elaborate significance of the current finding in industrial production of graphene layers.

Authors reply:

Although we acknowledge the importance of the industrial production of graphene layers, we do not consider it to be vital for attracting the interest of the broad readership of Nature Communications. In this opinion we rely on the Aims & Scope of the journal. The novel method of analytical chemistry reported in our work is not restricted to graphene-related topics but has much wider applicability and therefore will be interesting for the general audience of the journal.

2) Consistencies are not well-described between the present research and the report by Simonov et al [32].

i) The $C_{28}H_8/Cu(110)$ model by Simonov et al [32] explains three components in the C 1s core-level spectrum, shown in Fig.6 (b) in [32]. But the $C_{28}H_{14}/Cu(110)$ model cannot explain the experimental evidence since it should have only two components. Thus, the model sounds incorrect.

Authors reply:

We are pleased to read this comment, as it demonstrates the common overrating of XPS when it comes to identifying chemical species. Once more, this highlights the significance of our manuscript and its timeliness. Curve fitting of energetically unresolved features in XPS is always problematic and model-dependent. As outlined below, we believe that the Simonov et al. data (and our own XPS results for this system) can be fitted equally well with either two or three components. This is demonstrated for our XPS data in Authors Reply Figure R2. XPS is thus not conclusive on this system.

Indeed, in their work Simonov et al. have successfully used XPS to prove that a reaction of DBBA upon annealing on Cu(110) takes place, as the XPS spectra before and after annealing are evidently different (cf. Simonov et al. Fig. 6 (a) and (b)). But the interpretation of these spectral changes is far from clear:

1. Because Simonov et al. *a priori* assume a dehydrogenated state (cf. Simonov et al. Fig. 6 (b)), they fit the experimental spectrum including the shoulder with *three* components. However, in order to do this successfully, they have to shift the C1 central carbon peak

Authors Reply Figure R1. Fig. 6 from Simonov et al., Black and cyan vertical bars in (b) and (c) and red-framed inset in (b) are added to illustrate the authors' reply to Reviewer #1.

and the C2 hydrogenated carbon peak *differentially* (by approximately 100 meV and 200 meV, respectively, cf. Authors Reply Figure R1 (b)(c)) to higher binding energies compared to their reference spectrum of 7-AGNRs on Cu(111) (cf. Simonov et al. Fig. 6 (c)). The reason of the shift as such and its differential nature is not clarified by the authors. In fact, it would be hard to justify it.

2. A zoom into the experimental XPS spectrum of Simonov et al. at the shoulder clearly shows that the three-component fitting is in fact not very accurate (cf. Authors Reply Figure R1 (b) and the inset therein).
3. In order to check the stringency of the three-component model, we have carried out our own XPS study, and applied both *two* and *three*-component models to fit the XPS data. From Authors Reply Figure R2, it is clear that the experimental spectrum can be equally well fitted with both models.

In summary, this proves that in the present case XPS is not sufficient to reach conclusions about reaction intermediates in a model-free manner.

We stress once again that disproving the conclusions of Simonov et al. is **not** the main message of our manuscript. Therefore, we avoided a discussion of this issue in the original submission. Now, in view of the Reviewer's comment, we include an analysis of the XPS data in the Supplementary Information, although the conclusiveness of our photoemission tomography data, which are the main focus of our manuscript, actually makes this redundant.

Authors Reply Figure R2. Fitting of the XPS spectrum of the reaction intermediate measured at 500 eV excitation using (a) two- and (b) three-component models according to Simonov et al. Note that both models fit experimental data equally adequately. White dots represent experimental data. Black, cyan and red curves correspond to, respectively, C1, C2 and C4 components following Simonov et al., i.e. carbon atoms with three neighboring carbons (C1), hydrogenated carbon atoms (C2) and carbon atoms bonded to copper substrate (C4). Grey curves are the fitting envelop.

ii) In Fig.7 of [32], Simonov et al directly observed by STM the molecular orbital in real space that matches with $C_{28}H_{14}$ in vacuum and $C_{28}H_8/Cu(110)$. This indicates that shapes of the molecular orbitals are similar between $C_{28}H_{14}$ and $C_{28}H_8/Cu(110)$. On the other hand, simulated patterns of photoemission tomography are very different between molecules, as shown in Fig.4 in the present manuscript. To make the direct and fair comparisons with [32], images of the molecular orbitals must be added individually for nine tomography patterns in Fig.4.

Authors reply:

The Reviewer requests to include nine calculated real space density of states distributions (“*images of the molecular orbitals*”) for three orbitals and three possible reaction intermediates in question “*to make the direct and fair comparisons with [32]*”. Following the Reviewer’s request, we now include the corresponding simulated STM images for three interfaces $C_{28}H_{14}/Cu(110)$, $C_{28}H_{12}/Cu(110)$ and $C_{28}H_8/Cu(110)$ (cf. Supplementary Information of the revised manuscript and Authors Reply Figure R3). The simulations are based on density functional theory (DFT) calculations and are performed in the framework of the Tersoff-Hamann approximation. This is known to be a more correct approach for the interpretation of the STM images than comparing STM to “*images of the molecular orbitals*”.

It is clear that the simulated STM image corresponding to former LUMO of hydrogenated $C_{28}H_{14}/Cu(110)$ (cf. Authors Reply Figure R3 (d, 0.35 eV)) resembles the STM results of Simonov et al. (cf. Authors Reply Figure R4) much better than any of the images corresponding to the dehydrogenated $C_{28}H_8/Cu(110)$ (cf. Authors Reply Figure R3 (f)). This supports our conclusions from photoemission tomography.

Authors Reply Figure R3. (a-c) Chemical structure of intermediates and (d-f) simulated constant-current STM images for (d) $Br+C_{28}H_{14}/Cu(110)$, (e) $Br+C_{28}H_{12}/Cu(110)$ and (f) $Br+C_{28}H_8/Cu(110)$ interfaces. Binding energies used for simulations are noted in the insets. Color-code reflects the height of the tip (in Angstrom) above the surface.

We avoided discussing the STM contrast in our original submission to focus exclusively on the benefits provided by photoemission tomography. Now, in view of the Reviewer’s comments, we

include this in the Supplementary Information of the resubmitted manuscript.

3) The authors chose a photon energy of 35 eV for the experiment. Photon- energy dependence of the photoemission tomography should be mentioned. If the tomographic patterns vary with photon energy, choice of 35 eV should be described.

Authors reply:

We did not observe any significant change in the tomographic patterns of the reaction intermediate in the photon energy range from 20 to 40 eV. The chosen photon energy of 35 eV corresponds to the highest photoemission cross-section in the present experiment. Following the Reviewer's recommendation, we added a note in the *Methods* section.

Authors Reply Figure R4. Fig. 6a from Simonov et al.

Reviewer #2 (Remarks to the Author):

The present work by Yang et al. investigates the thermal induced reaction of DBBA on Cu(110) to graphene. The authors apply experimental and theoretical methods including photoemission tomography (PT) to identify a reaction intermediate that was previously possibly misinterpreted. The manuscript is well written, the findings are convincing and well presented, and the obtained results are novel and demonstrate the potential of PT to investigate dehydrogenation reactions that are often difficult to explore using other techniques. Therefore I believe that the present work is of great interest to the community of theoretical and experimental surface scientists and deserves being published in nature communications.

Authors reply:

We are pleased to read that the Reviewer finds our results interesting, well-presented and well-suited for the audience of Nature Communications.

I have only two points that should be addressed before publication:

-Why have the (orbital-)energies of the isolated molecule been calculated using the HSE06 xc-functional (Fig. 3c), whereas the band structure of the adsorbed molecules have been computed using the PBE functional (Fig. 4 d-f)? This inconsistency makes the claimed resemblance between Fig.3c and 4d less convincing and I would recommend to use the same functional in all systems.

Authors reply:

We can assure the Reviewer that for the molecule under study the frontier orbitals' order and shape are insensitive to the choice of the xc-functional. Actually, we have also performed the calculations for the adsorbed species with the hybrid HSE06 functional, however, for computational reasons with a rather coarse k-grid which leads to a poorly resolved k-map. This was the reason for presenting the results for the adsorbed species with the PBE functional. In order to avoid any confusion, in the revised manuscript we now show all computational results with the same PBE functional. Figure 3 and the *Methods* section are modified accordingly.

-In the abstract the authors say "we show that PT is extremely sensitive to the exact wave function". The present work employs KS-DFT orbitals to calculate momentum maps that are

compared to experimental maps. The term "exact wave function" is not appropriate and should be modified.

Authors reply:

We have modified this misleading statement to "Here we show that photoemission tomography is extremely sensitive to the character of the frontier orbitals."

Reviewer #3 (Remarks to the Author):

In their manuscript entitled 'Identifying surface reaction intermediates with photoemission tomography' the authors describe the comparison of experimental and simulated k-space patterns of molecular states as a method to identify reaction intermediates in a surface-chemical reaction. The manuscripts addresses the surface-assisted activation of dibromobianthryl (DBBA), which has a significant relevance in the field of bottom-up fabrication of graphene nanoribbons. The photoemission tomography data shown in the work is of high quality and is accompanied by a convincing simulation based on molecular conformations modeled by DFT.

Authors reply:

We appreciate the Reviewer's positive assessment of our experimental and theoretical results.

While the presented results give convincing new insights on a particular reaction step the chosen system (DBBA on Cu(110)), it also becomes clear that applicability of the method is limited to a very narrow selection of molecule-substrate combinations. For instance, DBBA was designed for yield 7-atom wide graphene nanoribbons on Au(111). This nanoribbons cannot be achieved on Cu(110) due to a fundamentally different reaction path on this surface. I am therefore afraid that the work cannot reach the wide readership at which Nature Communications is aiming. The insights gained here are from both sides, the presented method and the reaction investigated, too specific and more suitable for a specialized journal focusing on, for instance, surface-analytical aspects. On this basis I cannot recommend the manuscript for publication in Nature Communications.

Authors reply:

The Reviewer claims that "***The insights gained here are from both sides, the presented method and the reaction investigated, too specific and more suitable for a specialized journal.***" While the Reviewer provides some arguments that this reaction is specific for a particular type of nanoribbon, they do not provide any argument why our method should only be applicable to "***a very narrow selection of molecule-substrate combinations.***" We cannot possibly understand how the Reviewer reaches this conclusion. It must be a misunderstanding, because our method is based on the well-established experimental method of angle-resolved ultraviolet photoelectron spectroscopy, which can be applied to almost any sample. It is certainly more generally applicable than any type of orbital imaging by scanning tunneling microscopy.

Given the generality of our method, it evidently does not matter in which system we demonstrate its capabilities. We have chosen this particular reaction of DBBA, because the issue of dehydrogenation is difficult to tackle with other methods, such as XPS or STM. Therefore it illustrates the power of orbital tomography very convincingly, as the Reviewer concedes when stating "***the presented results give convincing new insights on a particular reaction step [of] the***

chosen system”.

We also note that the Reviewer is not entirely correct when they use the fact that nanoribbons cannot be grown on Cu(110) to conclude that the reaction path is fundamentally different from that on Cu(111). Indeed, while the substrate reactivity of Cu(110) prevents polymerization and graphene nanoribbons formation, the reaction path of DBBA to bisanthene and further to graphene on Cu(110) (cf. Simonov et al. Supporting Information) does include the reactions of dehalogenation and cyclodehydrogenation similar to the synthesis of graphene nanoribbons.

We again stress that the method presented in our work has a broader application and is not limited to the chosen exemplary case or to the synthesis of either graphene or graphene nanoribbons. Therefore we cannot agree with the Reviewer that the method of identification of the surface reaction intermediates presented in our work is too specific and cannot reach the wide readership of Nature Communications.

The authors also should consider the following points:

- The current discussion of the reaction mechanism assumes formation of bisanthene as an intermediate step. It is, however, not described how hydrogen passivation after dehalogenation would occur.

Authors reply:

Unlike formulated by the Reviewer, in our paper we *prove* (not “assume”) that the reaction intermediate is the fully hydrogenated bisanthene. This conclusion is based on pure experimental findings and their comparison with theory. We do not discuss the origin of hydrogen involved in the reaction and avoid corresponding speculations because it is irrelevant in the context of our manuscript. However, our chemical energy calculations clearly reveal the energetic profit of the passivation over the case of a chemical bonding to copper.

- The angle under which the EDCs shown in Fig. 2 were measured should be indicated.

Authors reply:

To obtain the corresponding energy distribution curves, the emission was integrated in the polar angle range from 0° (normal emission) to 85° (grazing emission) as explicitly written in the *Methods* section. This is now additionally noted in the figure caption.

- The caption of Fig. 2 does not specify the temperature to which the sample has been annealed.

Authors reply:

The annealing temperature (250°C) was actually specified in the title of the caption of Figure 2 of the originally submitted manuscript and additionally in the main text. It is now additionally noted in the text of the caption of Figure 2.

- Currently, the conditions for the applicability of the method are indicated only in a very indirect way (unique orientational alignment fo adsorbates). This should be improved.

Authors reply:

Photoemission tomography is *not* restricted to the cases of unique orientational alignment of adsorbates. If different adsorbate alignments coexist, the orientational resolution can be achieved either using photoemission microscopy (cf. J. Felner et al., J. Phys.: Condens. Matter **31**

(2019) 114003) or applying deconvolution (cf., e.g., B. Stadtmüller et al., EPL **100** (2012) 26008, B. Stadtmüller et al., Nat. Commun. **5** (2014) 3685).

Therefore, because photoemission tomography is based on angular-resolved ultraviolet photoelectron spectroscopy (ARUPS), its only restrictions are those that are typical for ARUPS, i.e. vacuum environment (note, however, the recent progress in near ambient pressure photoemission spectroscopy) and sufficient conductivity of the samples. To highlight this, we have modified the last paragraph of the *Discussion*.

List of corrections

in response to the Reviewer #1:

- The section *S3. Simulation of scanning tunneling microscopy contrast* is added to the Supplementary Information.
- The section *S4. X-ray photoemission spectroscopy of the reaction intermediate* is added to the Supplementary Information.
- In the *Methods* section of the manuscript a sentence is added which reads
“A photon energy of 35 eV was used, which corresponds to the highest photoemission cross-section in the present experiment.”

in response to the Reviewer #2:

- Figure 3 of the manuscript is modified.
- In the Abstract, a sentence is reformulated. Now it reads
“Here we show that photoemission tomography is extremely sensitive to the character of the frontier orbitals.”

in response to the Reviewer #3:

- Caption of Figure 2 is modified. The annealing temperature and photoemission intensity integration range is now included.
- In the *Discussion* section the sentence is added
“Its major restrictions are those of vacuum and conductive substrates inherent to ARUPS.”

REVIEWERS' COMMENTS:

Reviewer #1 (Remarks to the Author):

The revised paper now deserves publication.

There has been many papers that report a new technique and claim the previous results by classical methods without the detailed discussion. Giving an example of the research by Simonov et al. (ACS Nano 9 (2015) 8997), the authors faithfully discussed and added the information in the supplementary information. This comparison will help readers to understand appropriate applications of the new method.

Reviewer #3 (Remarks to the Author):

The authors have addressed the important questions arising from the first version of the submitted manuscript. The newly added information clearly supports the validity of photoemission tomography for identifying the conformation of critical intermediates in on-surface chemical reactions. All minor points were addressed and included in the manuscript. Although the authors could not convincingly show the general applicability of the method to a large range of molecule/substrate systems, I judge the further increased robustness of the presented assignment of intermediates for the chosen system to be a persuasive argument to publish the application of photoemission tomography for the identification of reaction intermediates.

I suggest publication of the new version in Nature Communications without further modifications.

Reviewer #4 (Remarks to the Author):

Conclusion

I recommend the publication of this paper in Nature Communication according to the following reason.

Short summary

I understood that the authors had taken an important step to show that the photoemission tomography (PT) combined with the theoretical simulation under the plane wave approximation for the final state of photoionization can be a powerful new tool to study the structure of surface reaction intermediate. Without doubt, innovative progress of science can be realized by developing a new experimental or theoretical tool, as we can understand from the history of science and engineering. I think that the present work meets also such a novelty as well as a novelty of the first accurate report on the structure determination of the reaction intermediate at the surface using DBBA on Cu(110) using the PT method.

Reason, reviewer's opinions and a few comments on minor points

The present research team succeeded to demonstrate that the imaging method of photoemission tomography (PT) combined with theoretical electronic structure computation can in principle offer direct and real-time information on the chemical structure of a reaction intermediate during the thermally induced reaction of molecules adsorbed on a solid surface using the DBBA-on-Cu(110) system. This method offers an intramolecular structure at room temperature and even at a higher temperature. This feature is a striking advantage of the PT comparing with STM and AFM that need a target specimen cooled down to lower temperatures for atomic resolution. The authors already wrote this point in the last part of the discussion section: <As photoemission tomography is neither constrained by the need for cryogenic temperatures nor to planar surface species, - - - -, we envision it to be a complementary companion to atomic force microscopy and other state-of-the-art surface science methods in the study of reaction pathways at surfaces.>.

As I will explain in the following paragraphs, the PT method can offer images (r -maps) of molecular orbitals (MOs) in molecular species on a solid surface at room temperature and even at higher temperatures by Fourier transform of observed k -map of photoelectron angular distributions, although MOs are not quantum-mechanical observables. It is clear that such measurements at room and higher temperatures are necessary to pursue a surface chemical reaction and observe the structure, the electronic states and corresponding MOs ($\psi(r)$) of reaction intermediates, but we have not yet reached the goal of having a completed powerful tool of this type (PT method).

The above-described incomplete realization of the PT method is related to the following problem. There is a drawback or limitation of the PT method because of a lack of evaluation of the validity of this method. Strictly speaking, the Fourier transforms, both of from the k -map (image on k_x - k_y plane) to the r -map (image on x - y plane, namely mapping of $\psi(r)$) and from the r -map to the k -map, are possible only when the measurements are performed at an appropriate photoelectron spectroscopy geometry, namely angular conditions of photoelectron excitation/detection. That is, the full angular-space photoelectron image (k_x - k_y maps) cannot be correctly simulated by the Fourier transform of the r -map of $|\psi(r)|^2$, because the Fourier transform needs to use the plane-

wave approximation for the final state of the photoexcitation. However, the effort of the present authors groups and Wuerzburg University group (Dr. A. Schoell & Prof. Reinert et al.) had reported step-by-step that accuracy and precision of the plane-wave approximation-based k-map simulation in the PT method are sufficient for a semi-quantitative discussion of the futures of both in the k- and the r-maps (refs.21-26) and for extracting information on the phase of $\psi(r)$ from $|\psi(r)|^2$ (ref.23). Such accumulation of results by the PT method may have been suggesting that evaluation of its accuracy and precision is essential to recognize it as a new powerful tool for obtaining/distinguishing $\psi_1(r)$ and $\psi_2(r)$ and the Fourier transforms [$\psi_1(k)$ and $\psi_2(k)$] under the plane wave approximation. Such a point somewhat resembles the reason why human eyes are more effective than the computer so far in distinguishing a cat face and a dog face in a poorly focused photograph. This type of function of human's eyes is similar to that realized in the recent success of the deep-learning method in AI technology using mathematical statistics in analyses of big data.

I understood that the authors took an important step in the present work to confirm a clearer advantage and potential of the PT method and to make it a powerful new tool to study the structure of surface reaction intermediate through presenting beautiful agreement between observed and simulated results of the "k-maps". They could then demonstrate the successful structure determination of the reaction intermediate in an atomic resolution, which is more precise and reliable than the STM results reported in ref [32].

I am personally interested in the comparison between the measurement-based and theory-based r-maps [$|\psi(r)|^2$ and $\psi(r)$], which are obtained by the Fourier transform of experimental results (k-map \rightarrow r-map), and computed $\psi(r)$, respectively. The authors will be able to return to a discussion using mathematical statistics on the accuracy and the precision of $|\psi(r)|^2$ and $\psi(r)$ obtained by the Fourier transform of an observed k-map. But this would be future work as the accuracy and the precision of the experimental $|\psi(r)|^2$ and $\psi(r)$ may be judged only by comparing with a correct theoretical $\psi(r)$, which depends on the accuracy and the precision of the theoretical method for electronic state calculation.

The present manuscript is well organized and easily understandable by readers from various fields.

From the above, I think that this paper by Yang et al. is worth publishing in Nature Communication. However, I have a few minor comments, which I want to ask the authors to consider, as given below.

Comments

Tiny request to the authors

I. Experimental information

The PT combined with theoretical computation/simulation is not straightforward to understand for non-experts of angle-resolved photoelectron spectroscopy. In particular it is not easy to understand the measurement geometry without figure (incidence photon angle, photon polarization direction,

the polar angle (θ) and azimuthal angle (ϕ) of the direction of the photoelectron, angular resolution ($\Delta\theta$ and $\Delta\phi$) and their relation to the toroidal electron analyzer. I therefore recommend the authors to give a simple figure of the angular geometry of the present measurement in Methods section or Supplementary Information. These angular geometries are also necessary to know the accuracy and the precision of the present PT results under the plane wave approximation.

II. Page 10: <Methods>

II-1. It is kind to have references to following {XX} parts:

(1) Metrology Light Source insertion device beamline {XX} of the Physikalisch-Technische Bundesanstalt (Berlin, Germany).

(2) the toroidal electron analyzer {xx}

II-2. Please describe so as not to misunderstand readers (indicated words with << >>).

(1) - - - - - was recorded with the toroidal electron analyzer <<in specular geometry>> over an emission - - -.

We thank all four Reviewers for reviewing our manuscript and supporting its publication in Nature Communications.

Reviewer #3 (Remarks to the Author):

The authors have addressed the important questions arising from the first version of the submitted manuscript. The newly added information clearly supports the validity of photoemission tomography for identifying the conformation of critical intermediates in on-surface chemical reactions. All minor points were addressed and included in the manuscript. Although the authors could not convincingly show the general applicability of the method to a large range of molecule/substrate systems, I judge the further increased robustness of the presented assignment of intermediates for the chosen system to be a persuading argument to publish the application of photoemission tomography for the identification of reaction intermediates. I suggest publication of the new version in Nature Communications without further modifications.

Authors reply:

The momentum-space imaging technique, i.e., kx-ky imaging of electronic states in reciprocal space, of the angle-resolved photoelectron spectroscopy has been successfully applied to a variety of molecular adsorbates including planar molecules in planar (cf., e.g., Ref. [1-4] and many other) and non-planar [5] adsorption configurations, nonplanar molecules [6], films with multiple symmetry domains [3,7,8], and polymerized extended structures, e.g. graphene nanoribbons [9]. It also has been applied to a broad variety of other systems; among others – two-dimensional materials (e.g., Ref. [10,11]), topological insulators (e.g., Ref. [12-14]), Weyl semimetals (e.g., Ref. [15-16]). Recently spin-resolved momentum-space imaging ARPES has been demonstrated [17,18]. Because the photoemission tomography technique used in our work for the chemical identification of the reaction intermediates is based on momentum-space imaging ARPES, we are strongly convinced that it is of general applicability to a broad range of molecule/substrate systems and even not limited to those.

1. P. Puschnig et al., Science 326, 702 (2009).
2. M. Wiessner et al., Nat. Commun. 5, 4156 (2014).
3. S. Weiss et al., Nat. Commun. 6, 8287 (2015).
4. G. Zamborlini et al., Nat. Commun. 8, 335 (2017).
5. E.M. Reinisch et al., Phys. Rev. B 93, 155438 (2016).
6. B. Stadtmüller (Technical University of Kaiserslautern, Germany), unpublished.
7. B. Stadtmüller et al., Nat. Commun. 5, 3685 (2014).
8. K. Schönauer et al., Phys. Rev. B 94, 205155 (2016).
9. B.V. Senkovskiy et al., 2D Materials 5, 035007 (2018).
10. S.Y. Zhou et al., Nat. Materials 6, 770 (2007).
11. E. Rotenberg et al., Nat. Materials 7, 259 (2008).
12. D. Hsieh et al., Science 323, 919 (2009).
13. Y. L. Chen et al., Science 325, 178 (2009).
14. M. Neupane et al., Nat. Commun. 4, 2991 (2013).
15. S.-Y. Xu et al., Science 349, 614 (2015).
16. L. X. Yang et al., Nat. Physics 11, 728 (2015).
17. C. Tusche et al., Nat. Commun. 9, 3727 (2018).
18. H.L. Meyerheim and C. Tusche, PSS-RRL 12, 1800078 (2018).

Reviewer #4 (Remarks to the Author):

Conclusion

I recommend the publication of this paper in Nature Communication according to the following reason.

Short summary

I understood that the authors had taken an important step to show that the photoemission tomography (PT) combined with the theoretical simulation under the plane wave approximation for the final state of photoionization can be a powerful new tool to study the structure of surface reaction intermediate. Without doubt, innovative progress of science can be realized by developing a new experimental or theoretical tool, as we can understand from the history of science and engineering. I think that the present work meets also such a novelty as well as a novelty of the first accurate report on the structure determination of the reaction intermediate at the surface using DBBA on Cu(110) using the PT method.

Reason, reviewer's opinions and a few comments on minor points

The present research team succeeded to demonstrate that the imaging method of photoemission tomography (PT) combined with theoretical electronic structure computation can in principle offer direct and real-time information on the chemical structure of a reaction intermediate during the thermally induced reaction of molecules adsorbed on a solid surface using the DBBA-on-Cu(110) system. This method offers an intramolecular structure at room temperature and even at a higher temperature. This feature is a striking advantage of the PT comparing with STM and AFM that need a target specimen cooled down to lower temperatures for atomic resolution. The authors already wrote this point in the last part of the discussion section: <As photoemission tomography is neither constrained by the need for cryogenic temperatures nor to planar surface species, - - - -, we envision it to be a complementary companion to atomic force microscopy and other state-of-the-art surface science methods in the study of reaction pathways at surfaces.>

As I will explain in the following paragraphs, the PT method can offer images (r-maps) of molecular orbitals (MOs) in molecular species on a solid surface at room temperature and even at higher temperatures by Fourier transform of observed k-map of photoelectron angular distributions, although MOs are not quantum-mechanical observables. It is clear that such measurements at room and higher temperatures are necessary to pursue a surface chemical reaction and observe the structure, the electronic states and corresponding MOs ($\psi(r)$) of reaction intermediates, but we have not yet reached the goal of having a completed powerful tool of this type (PT method).

The above-described incomplete realization of the PT method is related to the following problem. There is a drawback or limitation of the PT method because of a lack of evaluation of the validity of this method. Strictly speaking, the Fourier transforms, both of from the k-map (image on kx-ky plane) to the r-map (image on x-y plane, namely mapping of $\psi(r)$) and from the r-map to the k-map, are possible only when the measurements are performed at an appropriate photoelectron spectroscopy geometry, namely angular conditions of photoelectron excitation/detection. That is, the full angular-space photoelectron image (kx-ky maps) cannot be correctly simulated by the Fourier transform of the r-map of $|\psi(r)|^2$, because the Fourier transform needs to use the plane-wave approximation for the final state of the photoexcitation. However, the effort of the present authors groups and Wuerzburg University group (Dr. A. Schoell & Prof. Reinert et al.) had reported step-by-step that accuracy and precision of the plane-wave approximation-based k-map simulation in the PT method are sufficient for a semi-quantitative discussion of the futures of both in the k- and the r-maps (refs.21-26) and for extracting information on the phase of $\psi(r)$ from $|\psi(r)|^2$ (ref.23). Such accumulation of results by the PT method may have been suggesting that evaluation of its accuracy and precision is

essential to recognize it as a new powerful tool for obtaining/distinguishing $\psi_1(r)$ and $\psi_2(r)$ and the Fourier transforms [$\psi_1(k)$ and $\psi_2(k)$] under the plane wave approximation. Such a point somewhat resembles the reason why human eyes are more effective than the computer so far in distinguishing a cat face and a dog face in a poorly focused photograph. This type of function of human's eyes is similar to that realized in the recent success of the deep-learning method in AI technology using mathematical statistics in analyses of big data.

I understood that the authors took an important step in the present work to confirm a clearer advantage and potential of the PT method and to make it a powerful new tool to study the structure of surface reaction intermediate through presenting beautiful agreement between observed and simulated results of the "k-maps". They could then demonstrate the successful structure determination of the reaction intermediate in an atomic resolution, which is more precise and reliable than the STM results reported in ref [32].

I am personally interested in the comparison between the measurement-based and theory-based r-maps [$|\phi(r)|^2$ and $\phi(r)$], which are obtained by the Fourier transform of experimental results (k-map \rightarrow r-map), and computed $\phi(r)$, respectively. The authors will be able to return to a discussion using mathematical statistics on the accuracy and the precision of $|\psi(r)|^2$ and $\psi(r)$ obtained by the Fourier transform of an observed k-map. But this would be future work as the accuracy and the precision of the experimental $|\psi(r)|^2$ and $\psi(r)$ may be judged only by comparing with a correct theoretical $\psi(r)$, which depends on the accuracy and the precision of the theoretical method for electronic state calculation.

The present manuscript is well organized and easily understandable by readers from various fields.

From the above, I think that this paper by Yang et al. is worth publishing in Nature Communication. However, I have a few minor comments, which I want to ask the authors to consider, as given below.

Comments

Tiny request to the authors

I. Experimental information

The PT combined with theoretical computation/simulation is not straightforward to understand for non-experts of angle-resolved photoelectron spectroscopy. In particular it is not easy to understand the measurement geometry without figure (incidence photon angle, photon polarization direction, the polar angle (ϑ) and azimuthal angle (φ) of the direction of the photoelectron, angular resolution ($\Delta\vartheta$ and $\Delta\varphi$) and their relation to the toroidal electron analyzer. I therefore recommend the authors to give a simple figure of the angular geometry of the present measurement in Methods section or Supplementary Information. These angular geometries are also necessary to know the accuracy and the precision of the present PT results under the plane wave approximation.

II. Page 10: <Methods>

II-1. It is kind to have references to following {XX} parts:

(1) Metrology Light Source insertion device beamline {XX} of the Physikalisch-Technische Bundesanstalt (Berlin, Germany).

(2) the toroidal electron analyzer {xx}

II-2. Please describe so as not to misunderstand readers (indicated words with << >>).

(1) - - - - was recorded with the toroidal electron analyzer <> over an emission - - -.

Authors reply:

We are grateful to the Reviewer 4 for their detailed analysis our manuscript and a comprehensive comment and for recommending our paper for publication in Nature Communications. The Reviewer made the very valid request for the photoemission geometry in the toroidal analyzer experiments to, amongst other things, allow assessment of the plane wave approximation. This has now been made in the Supplementary Information along with a schematic of the experimental geometry.

To meet the Reviewer's requests,

I. in the revised version of the Supporting Information we add a section ***Supplementary Methods: Geometry of photoemission measurements*** and a figure ***Supplementary Figure 1: Schematic geometry of the photoemission experiment*** explaining the experimental geometry;

II-1. in the ***Methods*** section we included two references dedicated to the Metrology Light Source beamline and the toroidal analyzer;

II-2. we modified the sentence, to which the Reviewer is referring to.